# Factors associated with use of long-acting reversible and permanent contraceptives among married women in rural Kenya: A community-based cross-sectional study in Kisii and Kilifi counties

James Orwa[1,2☯]*, Samwel Maina Gatimu[3☯], Anthony Ngugi[2], Alfred Agwanda[1], Marleen Temmerman[1,4]

**1** Department of Public Health and Primary Care, Faculty of Medicine and Health Sciences, Ghent University, Ghent, Belgium, **2** Department of Population Health Sciences, Aga Khan University, Nairobi, Kenya, **3** School of Economics, University of Nairobi, Nairobi, Kenya, **4** Centre of Excellence for Women and Child Health, Aga Khan University, Nairobi, Kenya

☯ These authors contributed equally to this work.
* orwa.ariaro35@gmail.com

## Abstract

Long-acting and permanent contraceptive methods (LAPM) are effective and economical methods for delaying or limiting pregnancies, however they are not widely used. The Kenya government is promoting the use of modern methods of family planning through various mechanisms. This study aimed to determine the prevalence and factors associated with the use of LAPM among married women of reproductive age in targeted rural sub-counties of Kilifi and Kisii counties, Kenya. Baseline and end line Data from a program implemented on improving Access to Quality Care and Extending and Strengthening Health Systems (AQCESS) in Kilifi and Kisii counties of Kenya were used. Multi-stage sampling was used to sample 1117 and 1873 women for the end line and baseline surveys, respectively. Descriptive analysis was used to explore the respondents' characteristics and use of LAPM on a self-weighted samples. Univariable and multivariable binary logistic regression models using *svy* command were used to assess factors associated with the use of LAPM. A total of 762 and 531 women for the baseline and end line survey, respectively were included in this study. The prevalence of use of LAPM for baseline and end line survey were 21.5% (95% CI: 18.7–24.6%) and 23.2% (95% CI: 19.6%-27.0%), p-value = 0.485. The use of LAPM in Kisii and Kilifi counties was higher than the national average in both surveys. The multivariable analysis for the end line survey showed having 3–5 number of children ever born (aOR = 2.04; 95% CI: 1.24–3.36) and future fertility preference to have another child (aOR = 0.50; 95% CI: 0.26–0.96) were significantly associated with odds of LAPM use. The baseline showed that having at least secondary education (aOR = 1.93; 95%CI: 1.04–3.60), joint decision making about woman's own health (aOR = 2.08; 95%CI: 1.36–3.17), and intention to have another child in future (aOR = 0.59; 95%CI: 0.40–0.89) were significantly associated with the use of LAPM. Future fertility preference to have another child was significantly

**Data Availability Statement:** All relevant data are within the paper and its Supporting Information files.

**Funding:** The authors received no specific funding for this work.

**Competing interests:** The authors have declared that no competing interests exist.

associated with the use of LAPM in the two surveys. Continued health promotion and targeted media campaigns on the use of LAPM in rural areas with low socioeconomic status is needed in order to improve utilization of these methods. Programs involving men in decision making on partner's health including family planning in the rural areas should be encouraged.

## Background

Long-acting reversible and permanent contraceptives are one of the most effective and economical family planning methods worldwide to due to its numerous health benefits of preventing unintended pregnancies, promoting healthy birth spacing, reducing lifetime risk of maternal deaths, and enhancing attainments of development goals such as education and gender equality [1]. The World Health Organization (WHO) recommends spacing births for at least 24 months between pregnancies to give a mother time to recover from previous pregnancy since shorter birth intervals are characterized by negative maternal, foetal, and infant health outcomes [2, 3]. The use of contraceptives among women of reproductive age in low- and middle-income countries (LMICs) is still low compared to high-income countries [4]. Regions with high fertility rates tend to have a low contraceptive prevalence rate (CPR) and sub-Saharan Africa (SSA) region is not exceptional [5]. The CPR is defined as percentage of married or in-union women aged 15–49 years who are currently using, or whose sexual partner is currently using, at least a method of contraception.

Globally, there was an increase in CPR from 42% [6] in 1990 to 49% [7] in 2019 among women of reproductive age. In SSA, the CPR had increased from 13% to 33% between 1990 and 2020 [8], in Oceanic, from 20% to 28%; in Northern Africa and western Asia, from 26% to 34%, in central and southern Asia from 30% to 42% and in Latin America and the Caribbean from 40% to 58% [6]. In Kenya, the CPR increased from 33% in 1993 to 58% in 2014 [9]. Kenyan CPR in 2014 was lower than Rwanda (64.1%) in 2019, but higher than in neighbouring countries such as Burundi(28.5%) in 2016,Ethiopia (41.4%) in 2019, Tanzania (38.4%) in 2015, and Uganda (39.0%) in 2016 [9].

Family planning is a highly cost-effective intervention in Kenya with a healthcare cost saving of US $4.48 for every US $1 spent [10] and is one of the prioritized population and development programs [11]. In 1982, the National Council for Population and Development (NCPD) was established to "coordinate the implementation of population and development activities in Kenya," including mobilising political and financial support for population, family planning and reproductive health policies and programmes [12]. In addition, Kenya committed to the FP2020 goals to improve access to family planning services in order to increase modern CPR (mCPR) to 66% by 2030, adolescent women CPR to 55% by 2025 and to reduce teenage pregnancy from 18% to 10% by 2025 [13]. With these mechanisms to promote family planning programs, Kenya increased the CPR among married women from 7% in 1978 to 33% in 1993, 58% in 2014 [14], and 62.8% in 2019 in selected regions of the country [15].

Women can choose from a wide range of contraceptives, which include short-acting or long-acting reversible contraceptives (LARC) or permanent methods [16]. Most Kenyan women uses short-acting reversible contraceptives mainly injectables and pills, which have high discontinuation or switching rates compared to intrauterine contraceptive devices (IUCD) and implants [14, 17]. The high discontinuation rate of short-acting methods could be due to switching to more effective methods, contraceptive method stock-outs, prefer to conceive or fear of side effects associated with a particular method [14]. Contraceptive discontinuation among women with unmet need of contraception exposes a woman to the risk of

unintended pregnancy and induced abortions [18, 19]. Promoting the use of long-acting reversible and permanent methods (Intra-Uterine Devices (IUDs), Implants, Tubal ligation, female sterilization and Vasectomy) with low discontinuation rate could be beneficial to many of these women.

The 2014 Kenya demographic and health survey (DHS) estimated that only 9.9%, 3.4%, 3.2%, and 0.0% of married couples used implants, IUCD, female sterilization, and vasectomy, respectively [14]. The low national prevalence could be attributed to the availability and access of these methods as suggested by the proportion of health facilities in Kenya offering IUCD, implants, female, and male sterilization to be 75%, 58%, 7%, and 5%, respectively [20], or other factor related reason, opposition to use, lack of knowledge, and method related reasons; lack of trained providers and wide availability of short-acting methods in the rural areas [21]. Previous studies associated the use of LAPM with the number of living children [22, 23], having three or more children [22, 24–27], area of residence [22], region [22, 28, 29], woman's age [22, 23, 28, 29], education levels [23, 27, 28, 30–32], wealth status [23–25, 28, 31], joint decision making on family planning use with partner [33, 34], no desire for more children [30], and level of knowledge on LAPMs [26, 32]. While determinants of LAPM have been documents in several programs implemented in diverse settings to promote utilization, there remain a need to understand prevalence and determinants in the rural Kenyan context. Therefore, the study aimed to determine the prevalence and factors associated with the current use of long acting (Implant and IUCD) and permanent (Vasectomy and female sterilization)contraceptive methods among married women of reproductive age (15–49 years) in rural Kenya based on a community study in targeted areas of Kisii and Kilifi counties.

## Methods

### Data source, setting, and population

The study used data from the baseline and end-line survey of the AQCESS project, conducted between August and September 2016 and January and February 2020, respectively in four rural targeted implementation sub-counties of Kisii and Kilifi counties. The AQCESS project aimed to contribute to the reduction of maternal and under-five mortalities in Kenya; its organization and implementation have been described in our previous papers [35, 36]. The project promoted the use of family planning through community sensitisation messages by the community health volunteers (CHVs) as one of the implementation activities, however, there was no family planning commodity provided by the project during the implementation period. Details of design and conduct of baseline survey is discussed in our previous paper [35]; the below sections described the design and conduct of the end line survey.

The repeat cross-sectional survey was conducted in Kaloleni and Rabai sub-counties in Kilifi County and Bomachoge Borabu sub-county in Kisii County with a population of 304,778 and 129,617, respectively [36]. The maternal mortality rate in Kilifi was 448 deaths per 100,000 live births, and an under-five mortality rate of 87 deaths per 1,000 live births [36], with 70% of the population living below the poverty line. Under-five mortality rates in Kisii County was 36 deaths per 100,000 live births, however only 44% of its population living below the poverty line. Both counties have a high teenage pregnancy rate (Kisii 18.4% and Kilifi 21.8%) [36], which is higher than the national average of 18% [14, 36]. Kaloleni and Rabai sub-counties are served by 40 health facilities, while Bomachoge Borabu County is served by 12 health facilities [36]. About 82% and 93% of health facilities in Kilifi and Kisii offer family planning (FP) services; 49% and 93% of which offer services to adolescents, respectively. The CPR for the regions in 2014 were 34.1% and 66.1% for Kilifi and Kisii counties [20]. There are numerous structural challenges, including limited human healthcare resources, poor access to health

services due to geographical and transportation barriers, and limited healthcare infrastructure, including a high physician and nurse to population ratio in these counties [36].

The survey employed a two-stage sampling procedure considering a village as a cluster: In the first stage of sampling, a selection of 30 villages in each of the two sites were selected based on probability proportional to the number of households in each village. The second stage involved a random selection of households from a master frame of the household listing in the selected villages. The household listings were provided by the village elders who were familiar with the village boundaries. No further sampling was carried at the selected household level. All assented and consented women aged 15–49 years old and permanent residents of the selected villages were included in the survey.

The minimum sample size of 1788 households was calculated for the survey based on the following assumptions: an expected increase in the contraceptive prevalence rate between the baseline (2016) and end-line (2020) periods of 10%; 80% as the power; 95% level of significance; design effect of two; and a 20% non-response rate to account for absent household members during data collection. Figs 1 and 2 highlights the final study sample size for endline and baseline respectively after excluding women who were pregnant, unmarried, last menstruated more than six months before the survey, menopausal, never menstruated, did not know their last date of menstruation period, and those who preferred not to answer.

## Data collection tool and procedures

Trained enumerators collected data using tablets pre-installed with a standardized electronic questionnaire programmed on the Open Data Kit (ODK) software in English and the local languages of the study areas (Giriama, Ekegusi, and Swahili). The questionnaire adapted questions from the Demographic and Health Surveys and Multiple Indicator Cluster Surveys to allow for comparison. The questionnaire was pretested in the nearby villages which were not part of the actual survey before the start of the actual survey. Each supervisor was assigned team of six enumerators to manage; the supervisor ensured that the sampled households were interviewed and repeat visits conducted to absent household members conducted. After three unsuccessful repeat visits, a household was considered to have absent members for the duration of the survey. The collected data were synchronized into a secure, password-protected cloud-based server, allowing for real-time data quality assurance. Other quality checks performed during data collection included random spot-checks, close supervision of the enumerators, routine data cleaning, and addressing all identified issues prior to the start of data collection on any given day.

## Measures

The main outcome variable was LAPM use, which was assessed based on two questions: *(i)* *"Are you or your partner currently doing something or using any method to delay or avoid getting pregnant?"* and *(ii)* *"What are you currently doing to delay or avoid pregnancy?"*. Based on responses to the question (ii), women were classified as LAPM user (intrauterine contraceptive devices, implants, and male and female sterilization) or otherwise a non-user of LAPM. The non-users of LAPM includes users of short-acting/traditional methods (emergency contraception, injectable, male condoms, oral contraceptive pills, and traditional FP methods) and non-users of any method of contraceptive. Specifically the main outcome was binary in nature defined as:

$$Y_i = \begin{cases} 1, & User\ of\ LAMP \\ 0, & Otherwise \end{cases}$$

Where $Y_i$, is the response for the i*th* individual woman.

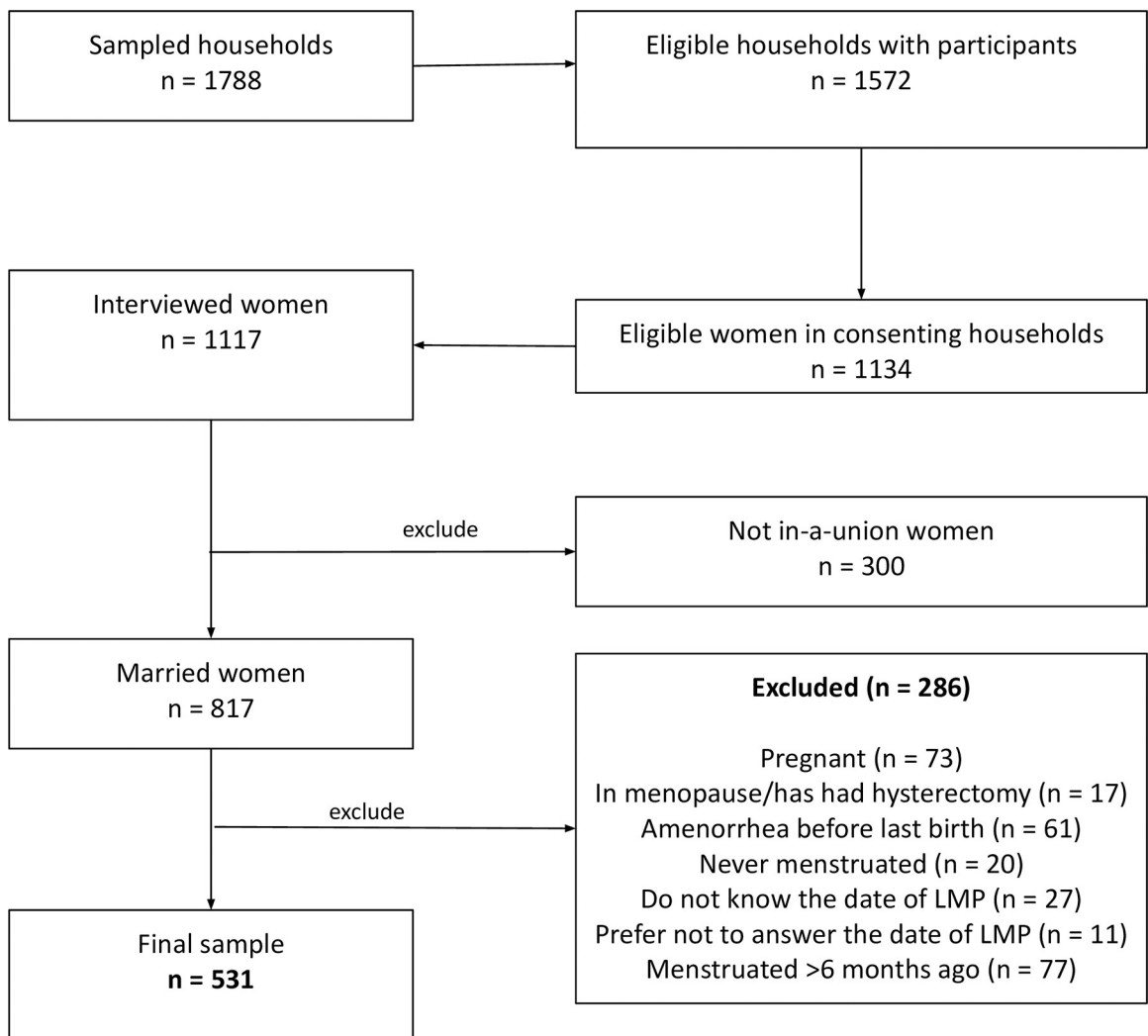

**Fig 1. The population of the women aged 15–49 years included and excluded in the endline analysis.**

Independent variables were identified based on a review of literature of previous studies on LAPM [19, 30, 31, 37–44] and included age measured in completed years at the time of the interview categorized into age groups (15–19, 20–29, 30–39, and 40–49 years); level of education (none, primary, and secondary and above), household wealth tercile (poor, middle, and rich), future fertility preference (no more/none and have ((an)other) child(ren), number of children ever born (CEB) as at the time of the survey (0–2, 3–5 and 6 or more), exposure to media was classified as yes for those who had listened to radio or watched television or read newspaper or accessed social media at least once a week and no for those who had not [23] and the main decision-maker on a woman's health (woman alone, woman with a partner, and other). The other decision-makers included the husband/partner alone, father or father-in-law, mother or mother-in-law, other male family members, and other female family members. Household wealth terciles were generated based on a wealth score computed using the principal component analysis approach using household assets and materials for the dwelling floor, roof, and external wall [45]. Variables on exposure to media and number of children ever born were not collected during the baseline survey.

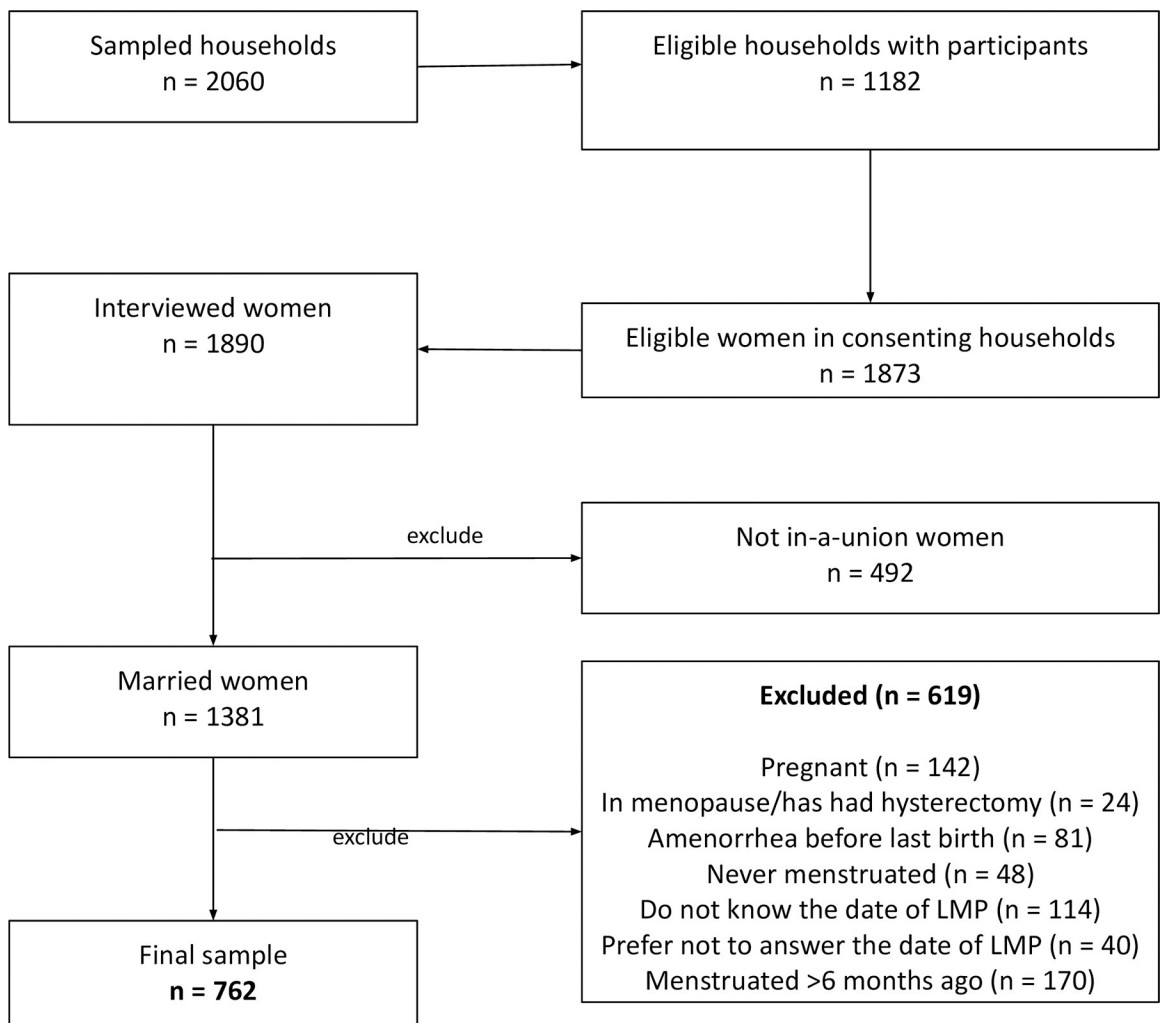

**Fig 2. The population of the women aged 15–49 years included and excluded in the baseline analysis.**

## Statistical analysis

Data were explored descriptively using the median and interquartile range (IQR) for continuous variables and frequencies and percentages for categorical variables. Chi-square test was used to compare the difference between the categorical groups. The multivariable logistic model was used to assess the factors associated with the use of LAPM while controlling for other variables. All the variable were considered to be important in explaining the outcome and were adjusted for in the multivariable logistic regression. The crude (cOR) and adjusted odds ratio (aOR) with the 95% confidence interval (CI) for the estimates is reported, and a p-value of less than 0.05 was considered statistically significant. We used "*svy*" set command in Stata to account for clustering due to complex sampling design of the study at village level. All analysis were conducted in Stata (15, StataCorp LLC, College Station, TX).

## Ethics

The study was approved by the Aga Khan University Institutional Ethics Review Committee and research permit provided by the National Commission for Science, Technology, and

Innovation. County approvals were obtained from the Ministries of Health in Kisii and Kilifi Counties. Written informed consent were obtained from each participant before the start of data collection, with those who could not write providing consent using their thumbprint. Women aged 15–17 years old provided assent and additional consent from their parents/ guardians or whoever they were comfortable with consenting on their behalf. The survey was conducted in line with the Helsinki Declaration on Research involving Human Subjects.

## Results

### Socio-demographic characteristics

Overall, 762 and 531 women were included in the baseline and end-line analysis, respectively. Majority of participants at baseline and endline were 30–39 years, poor, had completed primary school, and did not have or want more children (Table 1). In addition, at the end-line, more than two-thirds of the participants had been exposed to information on family planning in the media and majority had 3–5 children (Table 1). LAPM was used by 164 (21.5%), short-acting/traditional by 322 (42.3%) and 276 (36.2%) were not using any contraceptive during the baseline survey. In the endline survey, 123 (23.2%), 228 (42.9%), and 180 (33.9%) were using LAPM, short-acting/traditional and non-users respectively (Table 1).

### Prevalence of LAPM use

The prevalence of LAPM use were 21.5% (18.7%-24.6%) and 23.2% (95% CI: 19.6%–27.0%), p-value = 0.485 for the baseline and end line survey, respectively. Among the LAPM users during the endline, 24.5% used implants followed by female sterilization (7.1%), IUCD (2.8%) and male sterilization (0.6%). The use of these methods were slightly higher compared to the baseline (Fig 3). The proportion of women using LAPM was relatively high in Kisii County (55.3%) during the end line survey compared to baseline (43.3%). In the end line survey, LAPM prevalence was high among women were 30–39 years (44.7%), with primary level of education (49.6%), with 3–5 children (56.9%), and from the poor wealth tercile (77.2%) (Table 2). There was no significant difference in proportion in LAPM use between baseline and end line survey for different variables except in county of residence (p-value = 0.044), secondary and above level of education (p-value = 0.015), poor (p-value = 0.007) and rich (p-value = 0.018) wealth tercile, and decision making about own health with other family members (p-value = 0.035). (Table 2).

### Factors associated with the use of LAPM

In the baseline survey, county of residence, decision-making on own health, future fertility preference, and level of education were significantly associated with the use of LAPM. The odds of LAPM use were 51% less likely for women in Kisii compared to those in Kilifi (aOR: 0.49, 95% CI: 0.31–0.77) and were twice high likely for those women who jointly made decisions on their health with their partners compared to those who made decisions alone (aOR 2.08, 95% CI: 1.36–3.17). Women who preferred to have another child in future were 41% less likely to use LAPM compared to those who do not want a child in future (aOR 0.59, 95% CI: 0.40–0.89). Women with secondary level of education were 1.93 time high likely to use LAPM compared to women with no formal education (aOR: 1.93; 95%CI: 1.04–3.60) (Table 3).

In the endline, only future fertility preference and children ever born were significantly associated with LAPM use. Women with 3–5 children were two times more likely to utilize LAPM compared to those with 0–2 children(aOR 2.04, 95% CI: 1.24–3.36) whilst those who preferred another child in future had 50% reduced odds of LAPM use compared to those who did not have or want another child(aOR 0.50, 95% CI: 0.26–0.96) (Table 3).

**Table 1. Participants' characteristics stratified by county of residence at baseline and end line.**

| Variable | Baseline | | | End line | | |
|---|---|---|---|---|---|---|
| | | County of residence | | | County of residence | |
| | Overall | Kilifi, | Kisii, | Overall | Kilifi, | Kisii, |
| | N = 762 | N = 409 (53.7%) | N = 352 (49.2%) | N = 531 | N = 292 (55%) | N = 239 (45%) |
| **Age, years, Median (IQR)** | 31 (26–38) | 32 (26–38) | 30 (26–37) | 32 (26–38) | 32 (26–37) | 33 (27–38) |
| **Age-group, years, n (%)** | | | | | | |
| 15–19 | 25 (3.3) | 14 (3.4) | 11 (3.1) | 20 (3.8) | 7 (2.4) | 13 (5.4) |
| 20–29 | 292 (38.3) | 147 (35.9) | 145 (41.1) | 182 (34.3) | 111 (38) | 71 (29.7) |
| 30–39 | 296 (38.8) | 164 (40.1) | 132 (37.4) | 223 (42.0) | 115 (39.4) | 108 (45.2) |
| 40–49 | 149 (19.6) | 84 (20.5) | 65 (18.4) | 106 (20.0) | 59 (20.2) | 47 (19.7) |
| **Level of education, n (%)** | | | | | | |
| None | 131 (17.2) | 125 (30.6) | 6 (1.7) | 79 (14.9) | 79 (27.1) | 0 (0.0) |
| Primary | 420 (55.1) | 249 (60.9) | 171 (48.4) | 274 (51.6) | 168 (57.5) | 106 (44.4) |
| Secondary+ | 211 (27.7) | 35 (8.6) | 176 (49.9) | 178 (33.5) | 45 (15.4) | 133 (55.6) |
| **Wealth tercile, n (%)** | | | | | | |
| Poor | 492 (64.6) | 309 (75.6) | 183 (51.8) | 428 (80.6) | 238 (81.5) | 190 (79.5) |
| Middle | 143 (18.8) | 69 (16.9) | 74 (21.0) | 72 (13.6) | 39 (13.4) | 33 (13.8) |
| Rich | 127 (16.7) | 31 (7.6) | 96 (27.2) | 31 (5.8) | 15 (5.1) | 16 (6.7) |
| **Decision-making about own health, n (%)** | | | | | | |
| Woman alone | 185 (24.3) | 90 (22.0) | 95 (26.9) | 159 (29.9) | 70 (24.0) | 89 (37.2) |
| Other members* | 365 (47.9) | 233 (57.0) | 132 (37.4) | 123 (23.2) | 66 (22.6) | 57 (23.8) |
| Women jointly with partner | 212 (27.8) | 86 (21.0) | 126 (35.7) | 249 (46.9) | 156 (53.4) | 93 (38.9) |
| **Future fertility preference, n (%)** | | | | | | |
| No more/none | 411 (53.9) | 176 (43.0) | 235 (66.6) | 274 (51.6) | 112 (38.4) | 162 (67.8) |
| Have ((an) other) child(ren) | 351 (46.1) | 233 (57.0) | 118 (33.4) | 257 (48.4) | 180 (61.6) | 77 (32.2) |
| **Exposure to media, n (%)†** | | | | | | |
| Yes | | | | 364 (68.5) | 180 (61.6) | 184 (77.0) |
| No | | | | 167 (31.5) | 112 (38.4) | 55 (23.0) |
| **Number of children ever born n (%)†** | | | | | | |
| 0–2 | | | | 211 (39.7) | 109 (37.3) | 102 (42.7) |
| 3–5 | | | | 234 (44.1) | 129 (44.2) | 105 (43.9) |
| 6 or more | | | | 86 (16.2) | 54 (18.5) | 32 (13.4) |
| **Current use of contraceptives, n (%)** | | | | | | |
| LAPMs | 164 (21.5) | 93 (22.7) | 71 (20.1) | 123 (23.2) | 55 (18.8) | 68 (28.5) |
| Short/traditional | 322 (42.3) | 144 (35.2) | 178 (50.4) | 228 (42.9) | 119 (40.8) | 109 (45.6) |
| None | 276 (36.2) | 172 (42.1) | 104 (29.5) | 180 (33.9) | 118 (40.4) | 62 (25.9) |

†Data not collected during the baseline survey

* others included: husband/partner alone; mother or mother-in-law; other female family member

## Discussion

Slightly more than a third of married women in rural Kilifi and Kisii counties were using LAPM as at end line survey, which was higher than the national average of 28.5% [46], but was lower than the 57% of a community FP project in Western Kenya [47]. The improved use of contraceptives, including LAPM, could be attributed to ongoing efforts by the government and other health sector stakeholders towards promoting the uptake of modern contraceptive services in Kenya. The findings could also be attributed to the AQCESS project through community sensitisation messages by community health volunteers on the benefits of using family

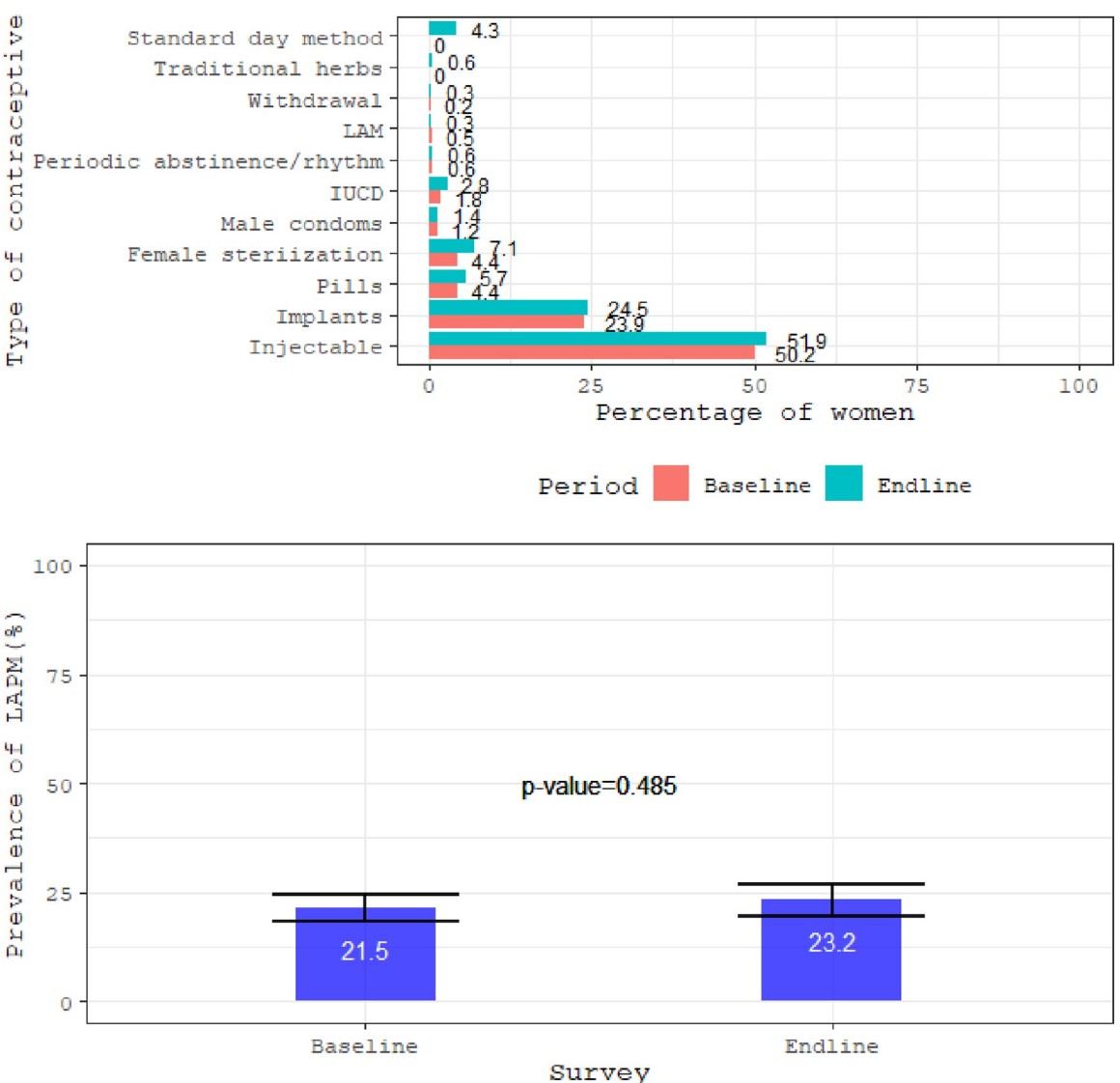

**Fig 3. Types of contraceptive current used by woman or husband/partner and prevalence of LAPM (baseline: N = 762 and endline: N = 531).**

planning. The gains on the use of LAPM in these areas could be sustained by the government and non-governmental organisations aimed at addressing provider- and health system-related barriers such as low numbers of trained providers [48, 49], recurrent stock-outs, and fear of side effects [50].

Injectable which is a short-acting and reversible contraceptives are the most preferred methods in the two surveys. Short-acting methods tend to be readily available in most of the rural public health facilities [20] at a low cost [51]. Their use may be associated with women in their early childbearing ages and those who would not wish to delay their pregnancies for long period of time of their reproductive career [52]. The study found implants and female sterilization as the most commonly used LAPM compared to IUCD and male sterilization. The findings were supported by findings in other settings [53, 54]. Once inserted, implants can last for two to five years and can be easily removed without any delay in return to fertility [55]. The

**Table 2. Prevalence of contraceptive use according to participants' characteristics at baseline and end line survey.**

| Variable | Baseline | | | End line | | | The difference in LAPM use between baseline and endline, p-value |
|---|---|---|---|---|---|---|---|
| | LAPM, N = 164 (21.5%)[1] | Short/ traditional, N = 322 (42.3%)[1] | None, N = 276 (32.2%)[1] | LAPM, N = 123 (23.2%)[1] | Short/ traditional, N = 228 (42.9%)[1] | None, N = 180 (33.9%)[1] | |
| **County of residence, n (%)** | | | | | | | |
| Kilifi | 93 (56.7) | 144 (44.7) | 172 (62.3) | 55 (44.7) | 119 (52.2) | 118 (65.6) | **0.044** |
| Kisii | 71 (43.3) | 178 (55.3) | 104 (37.7) | 68 (55.3) | 109 (47.8) | 62 (34.3) | **0.044** |
| **Age, years, Median (IQR)** | 30 (26–35) | 30 (25–36) | 34 (26–42) | 30 (26–35) | 32 (26–37) | 33 (28–41) | |
| **Age-group, years, n (%)** | | | | | | | |
| 15–19 | 5 (3.0) | 11 (3.4) | 9 (3.3) | 2 (1.6) | 10 (4.4) | 8 (4.4) | 0.439 |
| 20–29 | 65 (39.6) | 135 (41.9) | 92 (33.3) | 50 (40.7) | 80 (35.1) | 52 (28.9) | 0.862 |
| 30–39 | 73 (44.5) | 132 (41.0) | 91 (33.0) | 55 (44.7) | 100 (43.9) | 68 (37.8) | 0.973 |
| 40–49 | 21 (12.8) | 44 (13.7) | 84 (30.4) | 16 (13.0) | 38 (16.7) | 52 (28.9) | 0.959 |
| **Level of education, n (%)** | | | | | | | |
| None | 21 (12.8) | 35 (10.9) | 75 (27.2) | 9 (7.3) | 30 (13.2) | 40 (22.2) | 0.133 |
| Primary | 95 (57.9) | 184 (57.1) | 141 (51.1) | 61 (49.6) | 129 (56.6) | 84 (46.7) | 0.161 |
| Secondary+ | 48 (29.3) | 103 (32.0) | 60 (21.7) | 53 (43.1) | 69 (30.3) | 56 (31.1) | **0.015** |
| **Wealth tercile, n (%)** | | | | | | | |
| Poor | 102(62.2) | 203 (63.0) | 187 (67.8) | 95 (77.2) | 192 (84.2) | 141 (78.3) | **0.007** |
| Middle | 31 (18.9) | 66 (20.5) | 46 (16.7) | 17 (13.8) | 31 (13.6) | 24 (13.3) | 0.254 |
| Rich | 31 (18.9) | 53 (16.5) | 43 (15.6) | 11 (8.9) | 5 (2.2) | 15 (8.3) | **0.018** |
| **Decision-making about own health, n (%)** | | | | | | | |
| Woman alone | 34 (20.7) | 83 (25.8) | 68 (24.6) | 35 (28.5) | 62 (39.0) | 62 (39.0) | 0.130 |
| Other family members* | 62 (37.8) | 154 (47.8) | 149 (54.0) | 32 (26.0) | 52 (42.3) | 39 (31.7) | **0.035** |
| Women jointly with partner | 68 (41.5) | 85 (26.4) | 59 (21.4) | 56 (45.5) | 114 (45.8) | 79 (31.7) | 0.127 |
| **Future fertility preference, n (%)** | | | | | | | |
| No more/none | 95 (57.9) | 170 (52.8) | 146 (52.9) | 75 (61.0) | 111 (48.7) | 88 (48.9) | 0.603 |
| Have ((an) other) child(ren) | 69 (42.1) | 152 (47.2) | 130 (47.1) | 48 (39.0) | 117 (51.3) | 92 (51.1) | |
| **Exposure to media, n (%)** | | | | | | | |
| **Yes** | | | | 94 (76.4) | 164 (71.9) | 106 (58.9) | |
| **No** | | | | 29 (23.6) | 64 (28.1) | 74 (41.1) | |
| **Number of children ever-born, n (%)** | | | | | | | |
| 0–2 | | | | 40 (32.5) | 94 (41.2) | 77 (42.8) | |
| 3–5 | | | | 70 (56.9) | 102 (44.7) | 62 (34.4) | |
| 6 or more | | | | 13 (10.6) | 32 (14.0) | 41 (22.8) | |

**Bold**: Statistically significant at p < 0.05

**Table 3. Factors associated with the use of long-acting and permanent methods among women in rural areas, baseline (2016) and end line (2020) survey, Kenya.**

| Variables | Baseline | | End line | |
|---|---|---|---|---|
| | aOR (95% CI) | p-value | aOR (95% CI) | p-value |
| **County of residence** | | | | |
| Kisii | **0.49 (0.31–0.77)** | **0.004** | 1.09 (0.55–2.16) | 0.795 |
| **Age-group, years** | | | | |
| 20–29 | 1.00 (0.28–3.54) | 0.997 | 2.64 (0.45–15.53) | 0.263 |
| 30–39 | 0.92 (0.22–3.81) | 0.905 | 1.36 (0.22–8.43) | 0.725 |
| 40–49 | 0.40 (0.08–1.90) | 0.235 | 0.68 (0.09–5.43) | 0.702 |
| **Level of education** | | | | |
| Primary | 1.68 (0.94–2.98) | 0.076 | 2.01 (0.62–6.50) | 0.228 |
| Secondary+ | 1.93 (1.04–3.60) | **0.039** | 2.84 (0.68–11.83) | 0.139 |
| **Wealth tercile** | | | | |
| Middle income | 1.02 (0.59–1.74) | 0.144 | 0.86 (0.40–1.88) | 0.696 |
| Better income | 1.31 (0.90–1.89) | 0.949 | 1.68 (0.64–4.39) | 0.272 |
| **Decision-making about own health** | | | | |
| Women jointly with partner | **2.08 (1.36–3.17)** | **0.002** | 1.14 (0.60–2.19) | 0.671 |
| Other members | 0.83 (0.52–1.33) | 0.423 | 1.28 (0.79–2.09) | 0.299 |
| **Future fertility preference** | | | | |
| Have ((an) other) children | **0.59 (0.40–0.89)** | **0.013** | **0.50 (0.26–0.96)** | **0.040** |
| **Number of children ever born** | | | | |
| 3–5 | | | **2.04 (1.24–3.36)** | **0.008** |
| 6+ | | | 1.34 (0.51–3.51) | 0.530 |
| **Exposure to media** | | | | |
| Yes | | | 1.39 (0.79–2.46) | 0.233 |

aOR: adjusted odds ratio; CI: confidence interval; cOR: crude odds ratio; FP: family planning; IQR: interquartile range; LAPM: long-acting and permanent methods; Ref: reference category; significant factors in bold font.

method is most available rural public facilities in Kenya than IUCDs, female and male sterilization [20, 56]. Male and female sterilizations are only offered in 5% and 7% of health facilities, respectively [20] and are not widely accepted, especially in the rural setting. The finding is similar to a study in Uganda that showed that low uptake of these methods were associated with poverty, limited awareness of the method, lack of skilled personnel to administer the method, and limited resources to purchase and maintain laparoscopic equipment [57]. Poor knowledge and information about the methods, along with religious and socio-cultural barriers, are some of the perceived barriers to the uptake of this method [58]. This emphasises the need for continued awareness and family planning educational sessions on modern family planning methods, including LAPM, to clients and providers.

Women with at least secondary level of education had higher odds of using LAPM compared to those with no formal education. The possible explanation could be that educated women have increased access to information on the side effects, benefits of using LAPM, and of smaller family size. Increased educational attainment especially secondary school and above influence service use of and female decision-making power on reproduction health issues particular family planning. The findings agrees with other studies conducted in Kenya, Ethiopia and Uganda that had shown that higher education is an important predictor of LAPM use [23, 59, 60].

Number of children ever born had a mixed relationship with the use of LAPM, women with children between 3–5 showed positive significant increase on the use of LAPM, but at

higher number of children of 6 or more, there was a non-significant increase on the use of LAPM. This could be explained by the desire for women to have smaller family size and have decided to delay or limit future births LAPM; a method that offers long-term or permanent protection against unwanted pregnancies. This finding concurs with the finding of a study in Kenya and in Ethiopia [61, 62]. There are myths and misconception that these methods cause infertility which could be the reasons of low-use among women with 0–2 children. As the number of children increase fear of infertility related to those methods would decrease and women tend to use LAPMs [62]. On the other hand, women who wanted more children were less likely to use LAMP, which confirms the findings of a study in rural Kenya [30]. LAPMs offer efficient, long-term protection against pregnancy and women who may not have achieved their desired number of children may not prefer them as methods of choice [42].

## Limitations

The study was limited in assessing the direct impact of the AQCESS project on the utilization of LAPM. The project was on maternal and child health program with little activities on family planning. The increase shown in the uptake between baseline and endline could be an effort of other stakeholders. Second, the study was conducted in targeted areas in the two counties, and hence may not be generalizable to the whole country. Third, contraceptive use was self-reported, which may have resulted in an underestimation or overestimation of the prevalence of contraceptive use. There were also other variables that could have influenced the uptake of the LAPM services such as husband approval of the use of LAPM, myths and beliefs on the methods which were not assessed with these surveys. Lastly, the two studies were cross-sectional, and causal relationships could not be established; only possible associations between the outcomes and the explanatory factors were studied.

## Conclusion

About a third of married women in the study setting used LAPM, which is higher than the national average. The use of LAPM was associated with secondary and higher level of education, 3–5 number of children and future fertility preference. The study findings highlight a need for continued health promotion and media campaigns on family planning with a specific focus on the use of LAPM among women in rural areas and low socioeconomic status. Programs involving men in decision making about their health should be encouraged in these setting. Moreover, further research is needed to investigate the effects of other factors which were not studied in the current analysis, such as the use of media campaigns, training of health providers, equipping of health facilities, husband's approval on the use of LAPM, and religious and socio-cultural barriers on the utilisation of LAPM.

## Supporting information

**S1 File. English questionnaire for the survey.**
(DOCX)

**S1 Data. Anonymized dataset used in the analysis.**
(CSV)

## Acknowledgments

We acknowledge the Ministry of Health, the Kisii and Kilifi County governments, and the village elders for their support in conducting this study. We are grateful to the participants who

consented to the interviews and the AQCESS teams in Kenya (Kennedy Mulama, Rachel Odhiambo, Michaela Mantel, and Lucy Nyaga).

## Author Contributions

**Conceptualization:** James Orwa, Samwel Maina Gatimu, Alfred Agwanda, Marleen Temmerman.

**Data curation:** James Orwa.

**Formal analysis:** James Orwa.

**Methodology:** James Orwa, Samwel Maina Gatimu, Anthony Ngugi, Alfred Agwanda.

**Supervision:** Anthony Ngugi, Alfred Agwanda, Marleen Temmerman.

**Validation:** Marleen Temmerman.

**Visualization:** James Orwa, Samwel Maina Gatimu.

**Writing – original draft:** James Orwa, Samwel Maina Gatimu, Alfred Agwanda.

**Writing – review & editing:** James Orwa, Samwel Maina Gatimu, Anthony Ngugi, Alfred Agwanda, Marleen Temmerman.

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
