## [Decision Letter · Decision Letter 0]

3 Feb 2022

PONE-D-21-23920Long-acting and permanent contraceptives use and associated factors among married women in rural Kenya: a community-based cross-sectional studyPLOS ONE

Dear Dr. Orwa,

Thank you for submitting your manuscript to PLOS ONE. After careful consideration, we feel that it has merit but does not fully meet PLOS ONE’s publication criteria as it currently stands. Therefore, we invite you to submit a revised version of the manuscript that addresses the points raised during the review process.

We look forward to receiving your revised manuscript.

Kind regards,

Zelalem T. Haile, PhD

Academic Editor

PLOS ONE

Journal Requirements:

3. Thank you for stating the following financial disclosure: "The funders had no role in  analysis, decision to publish, or preparation of the manuscript"

"We are grateful to the Government of Canada, and the Aga Khan Foundation Canada for funding the end line survey from which data were extracted for this study. We acknowledge the Ministry of Health, and the  Kisii and Kilifi County governments, and the village elders, or support in conducting this study. We are grateful to participants who consented to the interviews and the AQCESS teams in Kenya (Kennedy Mulama, Rachel Odhiambo, Michaela Mantel, and Lucy Nyaga)."

"The funders had no role in  analysis, decision to publish, or preparation of the manuscript"

Additional Editor Comments (if provided):

Thank you for submitting your manuscript to PLOS ONE. After careful consideration, we feel that it has merit but does not fully meet PLOS ONE’s publication criteria as it currently stands. Therefore, we invite you to submit a revised version of the manuscript that addresses the points raised during the review process.

Reviewers' comments:

Reviewer's Responses to Questions

**Comments to the Author**

1. Is the manuscript technically sound, and do the data support the conclusions?

Reviewer #1: Yes

Reviewer #2: Partly

2. Has the statistical analysis been performed appropriately and rigorously? 

Reviewer #1: Yes

Reviewer #2: Yes

3. Have the authors made all data underlying the findings in their manuscript fully available?

Reviewer #1: No

Reviewer #2: No

4. Is the manuscript presented in an intelligible fashion and written in standard English?

Reviewer #1: Yes

Reviewer #2: Yes

5. Review Comments to the Author

Reviewer #1: Reviewer Comments

Reviewer – Wambui Kungu (PhD Demography)

Manuscript No. PONE-D-21-23920

Title - Long-acting and permanent contraceptives use and associated factors among married women in rural Kenya: a community-based cross-sectional study

Overall Comments;

The article addresses an important topic on the use of long-acting and permanent contraceptives

among married women in rural Kenya and the associated factors. The topic is important in Kenya for several reasons;

i) The use of effective contraceptives is important for women to prevent unintended pregnancies that might pose serious health and social complications such as unsafe abortions and maternal/child morbidity and mortality. Women will able to attain more in education and economic status if they can get children when they want.

For the country, to reduce high costs incurred in maternal/child health care, unwanted fertility as well as the high population growth rate.

ii) Increased use of these contraception which are more effective will address the challenge of unmet need for contraception in Kenya which is high 22%. Kenya is currently putting interventions towards achieving zero unmet need for contraception in line with its ICPD25 country commitments made in 2019 at the Nairobi Summit of the International Conference on Population and Development 25th Anniversary.

iii) More use of these contraceptive methods will help reduce contraceptive discontinuation which is high at 31% and waters down the high contraceptive prevalence rate the country has attained.

iv) It addresses the issue of contraception among the poor and marginalized

Setting: This is important because it is rural and hence the need for more exposure to contraceptive information and services for the women.

Methodology

Data- The data is original, collection methods and quality controls reported are important for credibility. Quality assurance is given

Model – The GEE regression model is appropriate for the population based survey which was using clustered data

Sampling- The sampling gives focus to inclusivity to avoid bias. Numbers of women interviewed and the sampling procedures are adequate- Exclusion criteria reported is important. Fig 1 shows the issues.

Results- They are well presented in statistical language and discussed but improvements can be done on the presented information in the tables

References- They are recent, relevant and therefore appropriate

Recommendation – The paper is of publishable quality but needs some revisions and English language editing.

Specific Comments on areas that can be improved

Presented as -Line No- Issue for review- Suggestion

1-3 Title – add Kisii and Kilifi at the end- This will create more interest in the study

Long-acting and permanent contraceptives use and associated factors among married women in rural Kenya: a community-based cross-sectional study in Kisii and Kilifi Counties.

30 Use regression model instead of approach - equations regression model….

41 Insert the word rural after targeting- LAPM targeting rural women

45 Rewrite the sentence for clarity- Contraception helps to space and limit the number of children and thus prevents unintended pregnancies and unsafe abortions

46 Add sometimes because it is not always the case that health problems arise -intervals are sometimes characterized….

53-57 Rewrite the sentence to shorten it for clarity

Line 56 –add mCPR after CPR

Line 56-change contraceptive to contraception--modern CPR (mCPR)….- any method of contraception…..

60/61 Improve clarity- long-acting reversible contraception (LARC) or permanent methods

64 Add Citation besides [15]- I propose add citation of KDHS 2014 which presents discontinuation data

66-68 Can add a sentence on what the Government direction is to back up the information-Refer to the National Family Planning Costed Implementation Plan, 2017-2020

83 Addition to sentence -rural Kenya based on a community study in Kisii and Kilifi Counties.

87-88 Sentence improvement -The counties may not be called rural but the sub counties can be rural- February 2020 in four rural sub counties of Kisii and Kilifi Counties

97 Correction on the teen pregnancy rates -Kisii is 18.4 and Kilifi 21.8-Citation [23]- 18-22%

Add national average of 18% -Reference(KDHS,2014)

109-112 The sentence needs rewriting for more clarity -Especially add something on the issue of 80% power and 20% increase, design effect of two

181-183 Mention age of youngest and oldest also

189 Table 1 -Change word characteristic to Variable

Sample characteristic to Number/%

201 Figure 2 - Change word current to currently

202-203 Table 2

The bivariate and multivariate presentation can be improved to avoid congestion which is not friendly to a reader Change word characteristic to Variable

Word multivariate may be better than multivariable

Maybe subdivide each into two

cOR 95% CI

aOR 95% CI

207 Multivariable analysis-Word multivariate may sound better than multivariable

208 Rewriting - The odds of using LAPM were ……

213 Tense- have ever- Women who had given birth…..

214 Rewriting/tense odds of using LAPM compared to those who had…

217 Rewriting- add found in-lower than 57% found in a community FP

220 Rewriting- add modern -uptake of modern contraceptive services

223 Rewriting- add organizations-non-governmental organizations targeted…

224 Make plural - such as low numbers of trained providers

228 Cut sentence for clarity- at a low cost [36]. They may be

230 Improve clarity- And in conformity to previous studies

232 Improve clarity- remove when needed- easily removed without…..

233 Improve clarity- implants more than IUCDs [20, 40].

235 Remove our- in the study setting

240 Improve clarity- about the use of permanent methods because they are irreversible and provider hesitancy to remove LARCs [34].

241-243 Rewrite sentence for clarity

245 Improve clarity- higher odds of using LAPM among

247 Improve clarity- explain their higher use of implants

248-250 Improve clarity- The study also found that women in the richest wealth tercile were more likely to use LAPM than women in the poorest wealth tercile, which mirrors findings in other studies [44, 45].

255/256 Improve clarity- settings were rural with poor geographical accessibility to health facilities and with limited human healthcare resources

257 Add more - with 3–5 children were more likely to use LAPM

258-260 Rewrite sentence for clarity - LAPM offers efficient, long-term protection against pregnancy, and that these women may have achieved their desired number of children, or want to space their births.

260-261 Average children is 3.9 (KDHS,2014)- Improve clarity - Besides, Kenyan women have on average 3.9 children ever born [11], which is consistent with the study findings.

269 Improve clarity- only possible associations

274 Improve clarity- omit and-ever tercile, those who did not want more children and those had given birth to 3–5 children.

276-277 Rewrite sentence for clarity - LAPM among women in rural areas and low socioeconomic status. Moreover, further research is needed to investigate the effects of other factors which…..

References

327 KDHS year of publication- 2015

342 Ref 16 - i)Check authors names are written correctly - ii)DHS Analytical Studies 20

362 Ref 21- Complete Report No.

Reviewer #2: PONE-D-21-23920

Long-acting and permanent contraceptives use and associated factors among married women in rural Kenya: a community-based cross-sectional study

Thanks for the opportunity to review this manuscript. The paper discusses a very relevant topic in low- and middle-income countries (LMICs) where contraceptive use, especially use of LAPM remains poor despite many years of investment on family planning programmes. While millions of women across LMICs would like to space or limit their number of children, non-use of contraceptives remains high among them despite their sexual exposure and an expressed intention to avoid pregnancy. The authors use endline data from a health programme to assess prevalence and factors associated with the use of LAPM among married women in rural Kenya. I have some observations which if addressed would improve the paper.

TITLE:

LN 1-3: ‘Long-acting and permanent contraceptives use and associated factors among married women in rural Kenya: a community-based cross-sectional study’ Consider changing the title to ‘ Factors associated with use of long-acting and permanent methods among married women in rural Kenya: a community-based cross-sectional study’

ABSTRACT:

Include the sample size

BACKGROUND:

Generally, the background of this paper should be strengthened to make clear the new knowledge gap the study is filling. There are numerous studies on factors associated with use of LAPM including some from sub-Saharan Africa which authors should consider highlighting.

LN 46: Authors wrote ‘Shorter birth intervals are characterized by maternal, foetal, and

47 infant health problems’…There need to link this sentence with the first so that it is not left hanging. Also, consider replacing ‘health problems’ with’ negative health outcomes

LN 49: I would recommend to the authors to define what CPR means for readers not well versed in the field

LN 54: indicate the mandate of NCPD when it comes to FP?

 

METHODS:

The study is based on endline data from a program on improving Access to Quality Care and Extending and Strengthening Health Systems in Kilifi and Kisii counties of Kenya. Given that this programme also promoted FP, the results could be biased unless the analysis controlled for participation. There is need to describe efforts to address this particular bias. Alternatively, authors should consider using baseline data instead. Is it possible to indicate the number of participants who adopted a method including LAPM between baseline and endline?

Include a description of the study design under the method’s section. If cross-sectional, was it a repeated cross-sectional design?

The outcome variable ‘use of LAPM’ is dichotomous so logistic regression would be preferable.

It is not clear why the authors included ‘Heard about FP on social media’ as one of the independent variables. It appears superfluous as it does not have any influence in using LAPM relative to non-LAMP. It is so missing from the list of IV listed in LN140-150

RESULT:

Table 1- column 2, the percentage for each variable should add up to 100%

Ln 193: Authors wrote ‘The prevalence of LAPM use among women using contraceptives….’ Should read ‘The prevalence of LAPM use among married women using contraceptives… The result should be limited to only married women.

LN 194: Use of implants (24.5%) and female sterilization (7.1%) is more than double what was reported in the last KDHS. We need to see more in the discussion on what could be contributing to such a high increase in the study settings

LN 202: Table 2. It is interesting that some of the variables are insignificant on bivariate model but significant on the multivariate model. For example, age, future fertility preference and CEB. This is problematic and makes interpretation open to doubt. There could be confounding and collinearity issues that authors should address.

DISCUSSION:

LN 220-222, Authors wrote ‘The findings could also be attributed to the possible indirect effects of the AQCESS project through community activities that encouraged FP utilization’ As already mentioned above, the intervention could bias the findings, therefore there need to control for participation. What elements of FP did the AQCESS project promote Did it only focus on LAPM or other methods as well? If both, then we expect an increase in the use of both LAPM and non-LAPM methods.

LN 238-243: In addition to demand- and supply-side barriers mentioned, there is also provider-related bias which may influence contraceptive uptake. Partner/Husband’s approval of the methods also play a big role especially in patriarchal societies where men dominate everything including reproductive health decisions.

LN 245: Authors wrote ‘ The higher odds of LAPM use among younger women may be attributed to their need to avoid unintended pregnancies and optimize birth spacing compared to older women. I find it difficult to accept this given that on the bivariate model, the association between age and use of LAPM is not significant. Generally, we expect use of LAPM to increase by age.

OVERALL COMMENT:

The article should be checked for grammatical errors throughout

---

## [Author Response · Author response to Decision Letter 0]

8 Apr 2022

see response to the reviewers attached with other documents

---

## [Decision Letter · Decision Letter 1]

2 Aug 2022

PONE-D-21-23920R1Factors associated with use of long-acting reversible and permanent contraceptives among married women in rural Kenya: a community-based cross-sectional study in Kisii and Kilifi countiesPLOS ONE

Dear Dr. Orwa,

Thank you for submitting your revised manuscript to PLOS ONE. After careful consideration, we feel that it has merit but does not fully meet PLOS ONE’s publication criteria as it currently stands. Therefore, we invite you to submit a revised version of the manuscript that addresses the points raised during the review process.

Specifically, we noticed that Reviewer #2 suggested a few important revisions that need to be addressed before publication. See below for their full comments.

We look forward to receiving your revised manuscript.

Kind regards,

Joseph Donlan, Senior Editor, on behalf of:

Bijaya Kumar Padhi, PhD, MPH

Academic Editor

PLOS ONE

Reviewers' comments:

Reviewer's Responses to Questions

**Comments to the Author**

1. If the authors have adequately addressed your comments raised in a previous round of review and you feel that this manuscript is now acceptable for publication, you may indicate that here to bypass the “Comments to the Author” section, enter your conflict of interest statement in the “Confidential to Editor” section, and submit your "Accept" recommendation.

Reviewer #1: All comments have been addressed

Reviewer #2: All comments have been addressed

2. Is the manuscript technically sound, and do the data support the conclusions?

Reviewer #1: (No Response)

Reviewer #2: Partly

3. Has the statistical analysis been performed appropriately and rigorously? 

Reviewer #1: (No Response)

Reviewer #2: Yes

4. Have the authors made all data underlying the findings in their manuscript fully available?

Reviewer #1: (No Response)

Reviewer #2: Yes

5. Is the manuscript presented in an intelligible fashion and written in standard English?

Reviewer #1: (No Response)

Reviewer #2: No

6. Review Comments to the Author

Reviewer #1: (No Response)

Reviewer #2: Thank you for the opportunity to review this manuscript for the second time. While the authors addressed some issues raised in the first draft, some still need attention, particularly on the methodology. See below my main comments:

General comments:

The articles have improved but need to be reviewed further for language and grammatical mistakes.

Introduction:

Line 46: What goals are you referring to here? Please indicate because the goals could be many.

Line 47: Insert the article 'The' before World and REPLACE 'World health organization' with 'World Health Organization. Also, add the acronym in the bracket, i.e., (WHO)

Line: 51-56: Authors wrote that 'Globally, regions with low fertility rates tend to have a high contraceptive prevalence rate (CPR) compared to sub-Saharan Africa (SSA), where the CPR is still low (5)'. Please add the CPR prevalence to show the magnitude of the variation you refer to. Compare CPR in the following order, global, regional, and national.

Line 68: indicate why we have a high discontinuation rate for injectables and Pill. I think discontinuation rates are high for the hormonal methods including implants, due to health-related reasons, including side effects and erroneous perception that some methods cause infertility.

Line 73: indicate which methods constitute the LAPM methods (i.e., Intra-Uterine Devices (IUDs), Implants, Tubal ligation, and Vasectomy) and why they have become a strategy for reducing maternal morbidity and mortality.

Methods:

Line 105-117: This paragraph highlights the context of the study, but it is heavy on MNCH indicators. Consider adding FP indicators that are already available from 2014 KDHS.

Measurement:

The study has some methodological biases that should be addressed. I still believe this study would add value by comparing the baseline and endline data sets, even though the design was not longitudinal. As already pointed out in the first draft, using endline data without controlling for women's participation in the project creates biases in the result since the AQCESS project promoted the use of family planning through community sensitization by the community health volunteers (CHVs). The authors assertion that family planning was not one of the critical interventions of the project is wrong since CHV sensitized women to FP. There is a possibility that some of the women were reached with FP messages; others may have not. In line 226, the authors allude to the effects of the AQCESS project on improving uptake of LAPM, but the analysis does not inform that.

While it is okay to exclude from the analysis menopausal women as well as those who had a hysterectomy, it is not clear why the analysis excluded non-users and pregnant women. Remember, non-use is also a choice women make regarding contraceptive use. One way of addressing this is to group the outcome variable into three categories: using long-acting and permanent contraceptive methods (IUD, female sterilization, and implant), using other methods (short-acting and traditional), and not using any method, and applying a multinomial logistic regression model instead of binary logistic.

In the flow diagram showing the analysis sample, it appears the study collected data on the timing the respondent started using LAPM. The timing a woman started using LAPM should be controlled for in the analysis. Such a variable could also assuage some of the weaknesses highlighted above.

Line 169: specify the confounders controlled for in the analysis

Line 166-173: Did the analysis control for the clustering effects? It is very important because of the survey design used.

Line 229-232: Authors argue that some supply-side factors are barriers to LAPM use, but I did not see that in the analysis result.

7. PLOS authors have the option to publish the peer review history of their article (what does this mean?). If published, this will include your full peer review and any attached files.

Reviewer #1: **Yes: **Dr. Wambui Kungu

Reviewer #2: **Yes: **George Odwe

---

## [Author Response · Author response to Decision Letter 1]

19 Sep 2022

The response to reviewer attached with the rest of the documents

---

## [Editor Report · Decision Letter 2]

21 Sep 2022

Factors associated with use of long-acting reversible and permanent contraceptives among married women in rural Kenya: a community-based cross-sectional study in Kisii and Kilifi counties

PONE-D-21-23920R2

Dear Dr. Orwa,

We’re pleased to inform you that your manuscript has been judged scientifically suitable for publication and will be formally accepted for publication once it meets all outstanding technical requirements.

Kind regards,

Bijaya Kumar Padhi, PhD, MPH

Academic Editor

PLOS ONE
---

## [Editor Report · Acceptance letter]

22 Sep 2022

PONE-D-21-23920R2 

Factors associated with use of long-acting reversible and permanent contraceptives among married women in rural Kenya: a community-based cross-sectional study in Kisii and Kilifi counties 

Dear Dr. Orwa:

I'm pleased to inform you that your manuscript has been deemed suitable for publication in PLOS ONE. Congratulations! Your manuscript is now with our production department. 

Kind regards, 

on behalf of

Dr. Bijaya Kumar Padhi 

Academic Editor

PLOS ONE